

# Comprehensive characterization of the particulate IVOCs and SVOCs from heavy-duty diesel vehicles using two-dimensional gas chromatography time-of-flight mass spectrometry

Xiao He[1], Xuan Zheng[1*], Shaojun Zhang[2,3], Xuan Wang[4], Ting Chen[1], Xiao Zhang[2], Guanghan Huang[2], Yihuan Cao[2], Liqiang He[2], Xubing Cao[5], Yuan Cheng[5], Shuxiao Wang[2,3], Ye Wu[2,6*]

[1]College of Chemistry and Environmental Engineering, Shenzhen University, Shenzhen 518060, China

[2]School of Environment, State Key Joint Laboratory of Environment Simulation and Pollution Control, Tsinghua University, Beijing 100084, China

[3]State Environmental Protection Key Laboratory of Sources and Control of Air Pollution Complex, Beijing 100084, China

[4]School of Energy and Environment, City University of Hong Kong, Hong Kong SAR, China

[5]State Key Laboratory of Urban Water Resource and Environment, School of Environment, Harbin Institute of Technology, Harbin, 150090, China

[6]Beijing Laboratory of Environmental Frontiers Technologies, Beijing 100084, China

*Correspondence to*: Xuan Zheng (x-zheng11@szu.edu.cn), Ye Wu (ywu@mail.tsinghua.edu.cn)



**Abstract.**

Tailpipe emissions from three heavy-duty diesel vehicles (HDDVs), complying with varying emission standards and installed with diverse aftertreatment technologies, are collected at a certified chassis dynamometer laboratory. The HDDV-emitted intermediate-volatility and semi-volatile organic compound (I/SVOC) emission and the gas-particle partitioning of the I/SVOCs are investigated. Over four thousand compounds are identified and grouped into twenty-one categories. The dominant compound groups of particulate I/SVOCs are alkanes and phenolic compounds. For HDDVs without aftertreatment devices, i.e., diesel oxidation catalyst (DOC) and diesel particulate filter (DPF), the emitted I/SVOCs partition dramatically into the gas phase (accounting for ~ 93% of the total I/SVOC mass), with a few exceptional categories: hopane, 4-ring polycyclic aromatic hydrocarbons ($PAH_{4rings}$), and $PAH_{5rings}$. For HDDVs with DPF and DOC, the particulate fractions are reduced to a negligible level, i.e., less than 2%. Nevertheless, 50% of the total 2-ring PAH mass is detected in the particle phase, which is much higher than the high-molecular-weight PAHs, arising from the positive sampling artifact of quartz filter absorbing organic vapours. The positive sampling artifact of quartz filter absorbing organic vapours is clearly observed and uncertainties are discussed and quantified. Particulate I/SVOCs at low-speed, middle-speed, and high-speed phases are collected and analysed separately. EF distribution of the speciated OA on a two-dimensional volatility basis set (2D-VBS) space reveals that the fractions of OA with O:C (oxygen to carbon) ratio > 0.3 (0.4, 0.5) are 18.2% (11.5%, 9.5%), 23% (15.4%, 13.6%), and 29.1% (20.6%, 19.1%) at low-speed, middle-speed, and high-speed stages. The results help to resolve the complex organic mixtures and trace the evolution of OA.





## 1. Introduction


The chemical composition of fine particle (particulate matter with aerodynamic diameter less than 2.5 micrometre,
$PM_{2.5}$) varies both temporally and spatially. Unlike the inorganic portion that has been well studied, the
characterization of organic aerosol, which takes up a major faction of $PM_{2.5}$ mass is yet to achieve (He et al., 2018;
Kundu et al., 2010). Nevertheless, the elevated $PM_{2.5}$ concentrations have been widely recognized to be associated
with enhanced mortality by epidemiologic studies (Franklin et al., 2008; Tecer et al., 2008; Pope et al., 2004;
Dockery, 2001; Laden et al., 2006). For example, exposures to polycyclic aromatic hydrocarbons (PAHs) and the
derivatives through inhalation, ingestion, and dermal contact are associated with an increased risk of cancer
(Durant et al., 1996; Umbuzeiro et al., 2008).
Once emitted into the atmosphere, the volatile organic compounds (VOCs), intermediate-volatility and semi-
volatile organic compounds (I/SVOCs) are subject to sequences of chemical and physical evolutions to form
secondary organic aerosol (SOA) (Alier et al., 2013; Paasonen et al., 2016; Wang et al., 2006; Stewart et al., 2021a;
Stewart et al., 2021b). I/SVOCs span a wide range of volatility and partition dynamically between the gas and
particle phases (Alam et al., 2019; Presto et al., 2009). The term effective saturation concentration ($C^*$, $\mu g\ m^{-3}$) is
frequently used to categorize IVOCs ($10^3 < C^* < 10^6\ \mu g\ m^{-3}$), SVOCs ($10^{-1} < C^* < 10^3\ \mu g\ m^{-3}$), and low
volatility organic compounds LVOCs ($C^* < 10^{-1}\ \mu g\ m^{-3}$) (Gentner et al., 2012). Diesel vehicle exhaust has
contributed significantly to the emission of VOCs, IVOCs, SVOCs, and PM on both global and regional scales
(Huang et al., 2015; Liu et al., 2021; Ridley et al., 2018). The abundant emission of the precursors and the dynamic
interactions under atmospheric conditions impose significant impacts on climate change and huma health (Luo et
al., 2022; Poorfakhraei et al., 2017). In view of such importance, the quantitative characterization of the vehicular
organic components, spanning the whole volatility range, is highly needed. While on-road vehicle emitted VOCs
have been well speciated and accurately quantified, regardless of fuel type, vehicle type, ignition system, and
driving condition, the determination of IVOCs and SVOCs is far from adequate (Kawashima et al., 2006; Gentner
et al., 2009).
The accurate quantification of I/SVOCs, which composes of thousands of individual compounds, remains a great
challenge (Stewart et al., 2021b). They are frequently reported as a few compound categories and leave the
majority being unresolved complex mixtures (UCMs) (Qi et al., 2019; Zhao et al., 2014). For instance, alkanes
(including *n*-alkanes, *i*-alkanes, and cyclic alkanes) are found to be the dominant fraction in I/SVOCs, contributing
to over 60% of total mass, followed by oxygenated and aromatic species (Alam et al., 2019; Lu et al., 2018; He
et al., 2022). Crucial structural information, e.g., carbon skeletons and chemical active moieties, is notably missing.
The knowledge of structural information at molecular level helps to give a more comprehensive description of the
chemical evolution of I/SVOCs from mobile sources and better predict the SOA formation (Chen et al., 2019;
Kleindienst et al., 2012; He et al., 2020; Tkacik et al., 2012). Besides, the molecular level composition alters the
optical properties of the OA significantly (Li et al., 2018; Li et al., 2021; Harvey et al., 2016).
The gas-particle (g-p) partitioning of vehicle emitted I/SVOCs is determined by the mutual effects of intrinsic
nature of the organics, e.g., the liquid vapor pressure, and the environmental conditions, e.g., temperature, bulk
OA concentration, and heterogeneous reactions (Lu et al., 2018; Sitaras et al., 2004; Chen et al., 2010; Liu et al.,
2015). The scenarios of g-p partitioning of vehicle emissions are described by different vehicle types or driving
conditions, and limited compound categories are reported (Lu et al., 2018; May et al., 2013a, b). The lacking of
phase distribution by chemical speciation biases the SOA model prediction and hinders a full understanding of



chemical fate of vehicle emissions (Li et al., 2018; Grieshop et al., 2007). For example, Zhao et al. (2013) reported
the g-p partitioning of individual organic species using a thermal desorption aerosol gas chromatography (TAG)
instrument and found that contribution of oxygenated compounds to SOA can be substantially increased through
g-p partitioning. However, a comprehensive characterization of speciated g-p partition of vehicle emission is yet
to achieve.
Given such significant research gap, particulate I/SVOCs at ascending speed stages are collected and analysed
separately. We integrate the targeted and non-targeted analysis to speciate and quantify them. The emission
characteristics are explored, and the speciation-by-speciation g-p partitioning is fully addressed. We observe
unusual absorption of IVOC vapours to the sampling surface (i.e., quartz filter), and provide a systematic
discussion on the sampling artifact/bias on g-p partitioning equilibrium. Particulate I/SVOCs at ascending speed
stages are collected and analysed separately. The results clearly demonstrate that the state-of-the-art instruments
enable the characterization of the complex organic mixtures and help to trace the evolution of organic aerosol.

## 2. Materials and methods

### 2.1 Vehicles, driving cycles, and sampling

The tailpipe emissions from the three in-use HDDVs are collected at the China Automotive Technology &
Research Centre (CATARC) in Tianjin, China. The vehicles are selected to cover a range of aftertreatment
technologies. One HDDV (#1) is installed with selective catalytic reduction (SCR) system and two HDDVs (#2
and #3) are installed with SCR, diesel oxidation catalyst (DOC), and diesel particulate filter (DPF). The recruited
HDDVs are modelled in year of 2016, 2020, and 2020, respectively and the gross weight are 18.7 t, 25 t, and 25
t. Vehicle #1 meets with China IV national emission standard which was implemented back to 2010 and vehicles
#2 and #3 comply with China VI national emission standard which come into force in 2021.
For each HDDV, they were tested on a chassis dynamometer (AIP-ECDM 72H/2AXLE) and operated over the
China heavy-duty commercial vehicle test cycle for heavy trucks (CHTC-HT) cold-start and hot-start driving
conditions consecutively. CHTC-HT driving cycle (1800 s) simulates the driving conditions for heavy-duty
commercial vehicles in China and is divided into three segments: low-speed (phase one (P1), 342 s), middle-speed
(phase two (P2), 988 s), and high-speed (phase three (P3), 470 s). Prior to cold-start, each vehicle was pre-
conditioned overnight to cool the engine completely and the time slot between cold-start and hot-start was
approximately 10 min. Each test cycle was duplicated for three times.
Constant volume sampler (CVS) is equipped with the real-time gas analyser module (MEXA-7400HLE, HORIBA,
Japan) to monitor the transient concentration of CO and $CO_2$. An array of on-line and off-line instruments are
deployed to measure the heavy-duty vehicle exhaust in the gas and particle phases. Experimental conditions
including temperature, air flow, relative humidity, and pressure and inorganic and organic components are
monitored collocated. The details about the sample collection of gaseous I/SVOCs are described elsewhere (He
et al., 2022). The particulate I/SVOC collection procedures are given below. Tailpipe emissions from each HDDV
is drawn into the CVS system, simultaneously with ambient air which is filtered by high-efficiency particulate air
filter. The diluted diesel exhaust is then directed into the second dilution trunk (SDT), where diesel emitted
particles are further diluted before entering the PM sampler and being collected by quartz filters. On each test,
one 47 mm quartz filter (Grade QM-A, Whatman, UK) is loaded for particle collection. The quartz filters are pre-



baked overnight at 550 °C to remove any carbonaceous contamination. The particle sampling probe places at the
centre line of the first dilution truck (DT) and 10 times DT inner diameter downstream the emission pipeline to
guarantee thorough mixing. The air flows, temperature and humidity control, and dilution ratios within the whole
sampling system follow the stipulations of the China VI emission standard (2018). The average temperature in
the sampling train is 47 ± 5 °C. Field blank samples are collected collocated at the upstream of the emission
pipeline. The experiment diagram to collect gaseous and particulate emissions and the position of gas monitors
are shown in Figure S1.

### 2.2 Sample treatment and chemical analysis

A total of 36 filter samples plus 3 field blanks were collected and subjected to the determination of I/SVOCs. A
precious portion of 1 cm$^2$ (1 cm × 1 cm) was removed from the quartz filter and cut into strips before placing
into the thermal desorption (TD) tube. 2 $\mu$L deuterated internal standard (IS) mixing solution was spiked onto the
strips through a mild $N_2$ blow (CSLR, Markes International). The list of IS species is shown in the supporting
information (S1). The TD tubes were placed into an automated thermal desorption system (TD100-xr, Markes
International), which is connected to a two-dimensional gas chromatograph (GC × GC) (Agilent 7890B, Agilent
Technologies) coupled with a time-of-flight mass spectrometer (ToF-MS) (LECO Pegasus4D, LECO
Corporation).
The TD, GC × GC, and ToF-MS parameters are similar to those previously published for the measurement of
gaseous I/SVOCs (He et al., 2022). Briefly, the TD tubes are heated to 315 °C for 20 min where the I/SVOCs are
vaporized gradually and condense at the cold trap which is kept at 25 °C. Next, the trap is heated to 330 °C for 5
min and the re-concentrated compounds are purged into the GC column in a split ratio of 8.7:1. The first Rxi-5ms
capillary column (30 m × 0.25 mm × 0.25 $\mu$m, Restek) and the second Rxi-17Sil MS (0.75 m × 0.25 mm ×
0.25 $\mu$m, Restek) capillary column are installed to separate the analytes. A modulator is deployed to partition the
effluents from the 1$^{st}$ column into cryo-focused segments and inject them into the 2$^{nd}$ column, with a modulation
time slot of 4 s. The column flow is set at 1.3 mL min$^{-1}$ and GC oven initial temperature at 50 °C for 5 min,
increased to 300 °C at 5 °C min$^{-1}$, and held for another 5 min. The secondary oven and modulator temperature are
5 °C and 30 °C higher than the GC oven temperature, respectively. The complete run time is approximately 3900
s. The ToF-MS is conducted in electron impact positive (EI +) mode (70 eV) scanning over an m/z range of
35−550 amu. The ion source temperature is kept at 250 °C.

### 2.3 Data analysis

Particulate I/SVOCs are identified and quantified with their respective authentic standards or surrogates using the
three-step approach proposed in He et al. (2022). In short, within one GC × GC chromatogram, for the peaks of
which the authentic standards are available, they are accurately identified based on the retention time of respective
authentic standards and their mass spectrum and precisely quantified based on the constructed calibration curves.
The list of authentic standards is shown in Table S1. Nest, for the peaks of which the authentic standards are not
available, they are semi-identified by referring to the elusion sequences and extracting mass spectrum patterns via
a self-developed algorithm. The syntax is described in He et al. (2022). Third, for the peaks without clear mass
spectrum patterns, they are semi-identified by the physically nearest surrogate within the GC × GC chromatogram.





The surrogate is picked out by iterating through all the authentic standards using the self-developed data
processing program and comparing the first retention time ($RT_1$) and second retention time ($RT_2$) intervals.
Basically, thousands of peaks are identified and grouped into twenty-one categories. The classified particulate
I/SVOCs include alkane, alkene, cycloalkane, hopane, 2-ring PAHs, 3-ring PAHs, 4-ring $PAHs_{202}$, 4-ring
$PAHs_{228}$, 5-ring PAHs, biphenyl & acenaphthene, acid, phenol benzylic alcohol, aliphatic alcohol ether,
aliphatic ketone ester, benzylic ketone ester, Nitros, $C_2$ alkyl benzene, $C_3$ alkyl benzene, $C_4$ alkyl benzene, $C_5$
alkyl benzene, and $C_6$ alkyl benzene.
**2.4 Calculation of emission factors (EFs)**
Particulate I/SVOC EFs are determined using the following equation by assuming that the $CO_2$ and CO are the
dominant combustion products of diesel fuel.
$$EF = (\frac{\Delta I/SVOCs}{V_S} \times \frac{V_{cvs} \times 10^6}{\Delta[CO_2] \times M_C/M_{CO_2} + \Delta[CO] \times M_C/M_{CO}})w_C$$
where EF is the emission factor of particulate I/SVOCs (mg·kg·fuel[-1]); $\Delta I/SVOCs$ is the mass deposited on the
quartz filter in the CVS (mg), which is corrected for the background contamination measured on the field blanks
and column bleedings; $\Delta[CO_2]$ and $\Delta[CO]$ are the background-corrected $CO_2$ and CO masses (mg), respectively;
$M_{CO_2}$, $M_{CO}$, and $M_C$ are the molar weight of $CO_2$, CO, and C atom; $V_{cvs}$ and $V_S$ are the air flow monitored in the
CVS and particle sampling trunk (L min[-1]); $w_C$ is the mass fraction of carbon (0.86) in the diesel fuel.
**3. Results and discussion**
**3.1 Emission factors and the chemical speciation and of particulate I/SVOCs**
Figure 1 shows the speciated emission factor of the HDDV-emitted I/SVOCs in the particle phase. Generally,
over four thousand individual peaks are detected within different tailpipe samples and grouped into twenty-one
categories after the three-step data treatment procedure. The average HDDV-emitted particulate I/SVOCs EFs of
cold-start and hot-start driving cycles are 147.8 and 1.7 mg·kg·fuel[-1] for non-(DPF + DOC) vehicles, 1.6 and 0.9
mg·kg·fuel[-1] for (DPF + DOC) vehicles. Substantial removal effect of the aftertreatment devices is confirmed. A
category specified EFs for the non-(DOC + DPF) and (DOC + DPF) vehicles are shown in Table S2.
In general, alkane is the most abundant species, taking up 22-63% of the total particulate I/SVOCs followed by
2-ring PAHs (20-33%) and phenol benzylic alcohols (14-17%). The sum of the three categories accounts for more
than 75% of the total particulate I/SVOCs. The EF of alkane derived from non-(DOC + DPF) vehicles under the
cold-start condition averages to 92 mg·kg·fuel[-1], which is two orders of magnitude higher than that of hot-start
cycle and (DOC + DPF) vehicles, as illustrated by the grey squares in Figure 1. Alkene and cycloalkane show
commensurate EFs, with the average values of 2.4 and 1.8 mg·kg·fuel[-1] for cold-start and 0.04 and 0.05 mg·kg·fuel-
[1] for hot-start driving cycle for non-(DOC + DPF) vehicles, accounting for minor parts of the total particulate
I/SVOCs. The emission of the two species is further reduced after the installation of aftertreatment devices.
2-ring to 5-ring PAHs are frequently detected in particulate I/SVOCs, which is different from the gaseous
I/SVOCs that only 2-ring to 4-ring PAHs were observed (He et al., 2022). The averaged EFs of PAH subgroup
vary significantly. For example, for non-(DOC + DPF) vehicles operated under cold-start driving condition, 2-
ring PAHs are detected at abundant concentration of 33.8 mg·kg·fuel[-1], whereas 3-ring, 4-ring, and 5-ring PAHs



are detected at much less concentration of 1.5, 1.3, and 0.8 mg kg·fuel$^{-1}$, respectively. It was reported that 16
priority PAHs listed by the United States Environment Protection Agency accounted for a minor fraction of the
total PAH mass and the non-targeted analysis has highlighted the significance of the un-resolved PAHs (An et al.,
2022; Chen et al., 2022). The two isomers, biphenyl and acenaphthene, contribute least (less than 0.2%) within
the PAH subgroup, which is consistent with previous findings (Hazarika et al., 2019).
Oxygenated compounds, including phenol benzylic alcohols, aliphatic ketones, benzylic ketones, and acids are
routinely detected. The EFs sum up to over 7% of the total mass. Aliphatic alcohols are observed to be abundant
in the gas phase but not detectable in particulate I/SVOCs (He et al., 2022). The installation of DPF and DOC
reduces the emission of oxygenates by over 93-99%. For instance, the EF of benzylic alcohols of non-(DOC +
DPF) vehicles is 2.83 mg kg·fuel$^{-1}$ whereas that of (DOC + DPF) vehicles is 0.15 mg kg·fuel$^{-1}$.
The EF of Nitros is measured to be 0.4 mg kg·fuel$^{-1}$ on average, taking up of 1.6% of the total mass. The
installation of DPF and DOC reduces the emission of Nitros by over 95%, from 1.08 mg kg·fuel$^{-1}$ to 0.05 mg
kg·fuel$^{-1}$. Mono-aromatic compounds, which were measured to take up over 10% of the gaseous I/SVOCs, are
negligible constituents in the particle phase (He et al., 2022).





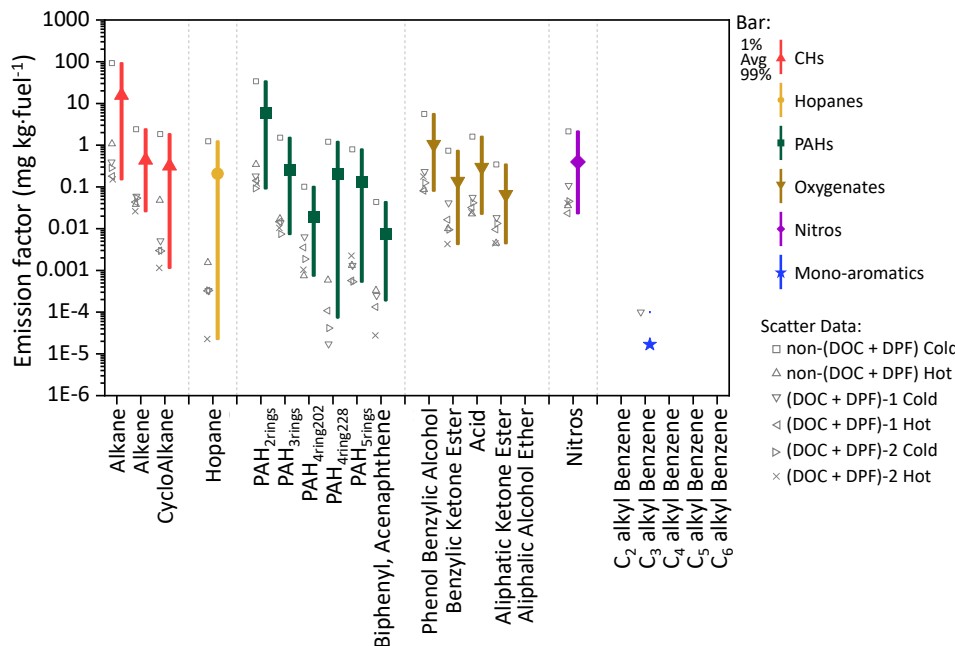


**Figure 1. The measured emission factor (mg kg·fuel$^{-1}$) of the twenty-one categories of the HDDV-emitted I/SVOCs in the particle phase. Coloured-bars and coloured-scatters/shaped-scatters represent different organic species and driving cycles. The square dots in the middle of each bar denote the average value and the lower and upper boundaries of the bar denote the 1% and 99% percentile of the values.**





**3.2 Volatility distribution of the speciated I/SVOCs and the comparison between cold and hot starts**

Figure 2 displays the volatility distribution of the speciated I/SVOCs under the cold-start and hot-start driving cycles for non-(DOC + DPF) vehicles and (DOC + DPF) vehicles. Inserted pie charts illustrate the color-labeled mass contributions of each compound category. The absolute values of I/SVOC EFs distributed in each volatility and O:C bin are summarized in Tables S3 - S8. The calculation of the saturation mass concentration is presented in the supporting information (S4). The volatility distributions among the four test conditions, i.e., cold-start non-(DOC + DPF) ($C_{wo}AT$), hot-start non-(DOC + DPF) ($H_{wo}AT$), cold-start (DOC + DPF) ($C_{wi}AT$), and hot-start (DOC + DPF) ($H_{wi}AT$), do not vary much except the two peaks at $\log_{10}C^* = -4 \, \mu g \, m^{-3}$ and $\log_{10}C^* = -3 \, \mu g \, m^{-3}$ under $C_{wo}AT$ and $H_{wo}AT$ (Figure 2a and 2b). The abnormal abundant emissions indicate intensive incomplete combustion processes, especially under cold-start condition. The high emissions at the low volatility end vanish after the installation of DOC and DPF (Figure 2a vs. Figure 2c, Figure 2b vs. Figure 2d), revealing that the aftertreatment devices eliminate the low volatility compounds, mostly alkanes, efficiently. Great environment benefits are thereby expected with the advancing of the aftertreatment technologies.

The majority of particulate I/SVOCs distribute in the volatility range of $\log_{10}C^* = 1$ to $8 \, \mu g \, m^{-3}$ while the specified compound categories distribute differently and could be classified into three groups. First, alkanes are observed within the whole volatility range at abundant level. Second, hopanes, $PAH_{4rings}$, and $PAH_{5rings}$ reside in the volatility range of $\log_{10}C^* \leq -2 \, \mu g \, m^{-3}$ dominantly. For example, about 44% of hopane mass are measured in the volatility bin of $\log_{10}C^* = -3 \, \mu g \, m^{-3}$. Third, light molecular weight PAHs, oxygenated compounds, and Nitros present in the volatility range of $\log_{10}C^* \geq 2 \, \mu g \, m^{-3}$ substantially. For example, phenol benzylic alcohols, the most abundant oxygenated compounds observed in particulate I/SVOCs, partition into the high volatile range entirely.

The mass fractions of O-I/SVOCs under the cold-start and hot-start driving cycles in the gas and particle phases are shown in Figure S3. The impacts of O-I/SVOCs on SOA formation are complex. On one hand, the formation potential of oxidized components is lower than that of hydrocarbons, for example alkane (Chacon-Madrid and Donahue, 2011; Donahue et al., 2011; Ziemann, 2011). On the other hand, the increasing O:C ratio adds fragmentation on the carbon skeleton which would facilitate SOA formation (Donahue et al., 2012; Kroll et al., 2009). An increasing trend of mass fraction of particulate O-I/SVOCs from low volatility end to high volatility end is clearly demonstrated whereas a bimodal pattern of gaseous O-I/SVOCs is observed. The gaseous O-I/SVOCs were divided into two major groups with one group peaking in the volatility range of $\log_{10}C^* = 4$ to $8 \, \mu g \, m^{-3}$ and another group prevailing in the volatility range of $\log_{10}C^* = -2$ to $3 \, \mu g \, m^{-3}$. The two groups possess different chemical structures and functional groups. They were fully addressed in previous work and will not be repeated here (He et al., 2022). By contrast, one compound category, phenol benzylic alcohols, dominates in the particulate O-I/SVOC. The mass fraction of phenol benzylic alcohols is 5% and 6% for non-(DOC + DPF) vehicles under cold-start and hot-start conditions, respectively. The mass ratio increases to 26% and 22% for (DOC + DPF) vehicles. It contributes significantly to the total mass in the high volatility range of $\log_{10}C^* = 6$ to $8 \, \mu g \, m^{-3}$.

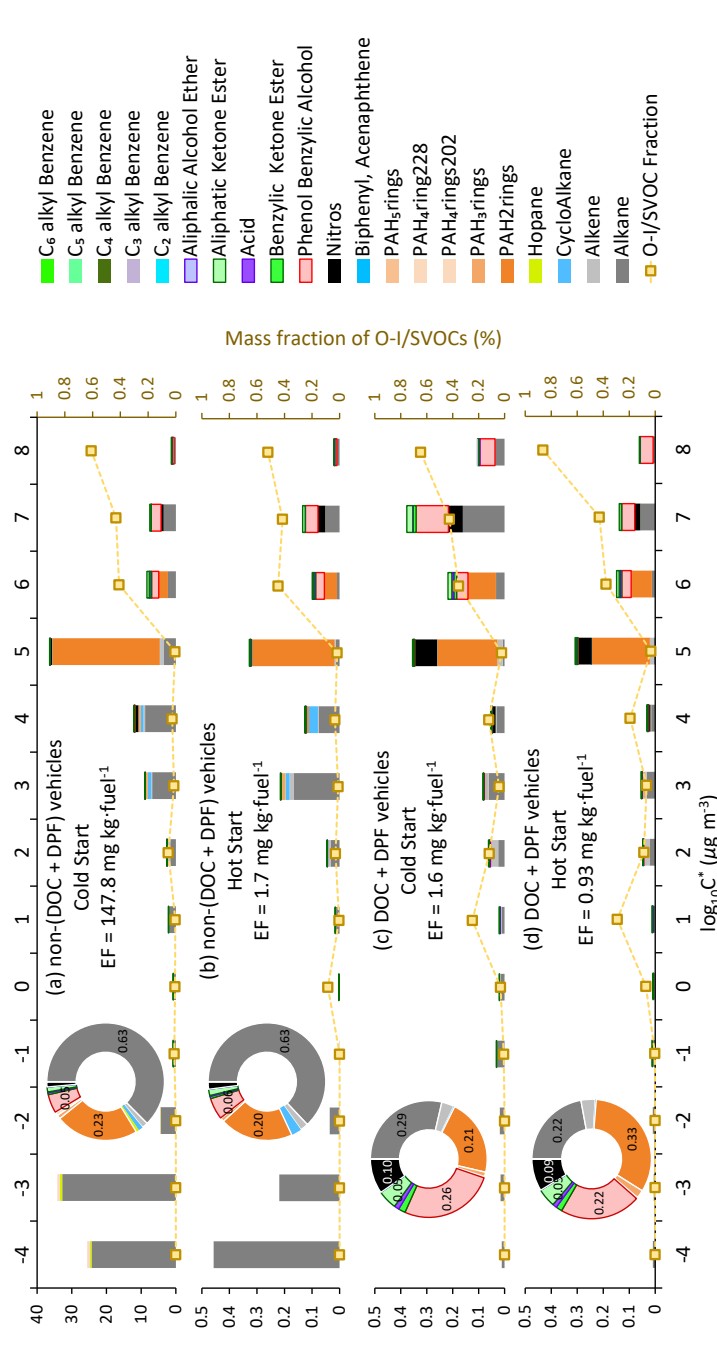

**Figure 2. EFs of particulate I/SVOCs under the cold-start and hot-start driving cycles. Different coloured bars represent different compound categories. Mass fraction of the O-I/SVOCs, indicated by the scattered squares, is scaled by the right axis. Embed pie charts are the mass fractions of different compound categories, and the numbers show the mass contributions of the top few species.**





**3.3 The EF distribution of particulate I/SVOCs**

Figure 3 and Figure 4 display the EF distribution of the speciated particulate I/SVOCs on a two-dimensional volatility basis set (2D-VBS) space of P1, P2, P3, and whole (W_cold and W_hot) driving cycles. The absolute values of I/SVOC EFs distributed in each volatility and O:C bin are summarized in Tables S3 – S12.

Distinct distribution patterns are observed between different speed stages. For non-(DPF + DOC) vehicles, peak signals of P1 are detected at low volatility and O:C ratio bins, i.e., $\log_{10}C^* = -3$ to $-4$ $\mu$g m$^{-3}$ and O:C < 0.3 whereas those of P2 and P3 are measured at $\log_{10}C^* = 3$ to 7 $\mu$g m$^{-3}$ and the fraction of I/SVOCs with higher O:C ratio increases, especially at high-speed stage (P3). For example, the fractions of I/SVOC with O:C > 0.3 (0.4, 0.5) are 18.4% (10.2%, 7.0%), 13.4% (8.3%, 7.6%), and 25.3% (19.4%, 18.6%) for P1, P2, and P3 stages. The fraction of I/SVOCs with higher O:C ratio decreases rapidly to less than 10% for low and middle-speed stages, contrast with which the fraction remains at comparable level for high-speed stage. The emission characteristics of the whole driving cycle combine the patterns of the separate speed phases, and a bimodal trend is observed as displayed in Figure 5d and 5e.

After the installation of aftertreatment devices, the peak signals of P1 are detected at high volatility bins, i.e., $\log_{10}C^* = 3$ to 7 $\mu$g m$^{-3}$ and low O:C range. In comparison with non-(DPF + DOC) vehicles, the EF volatility distribution of P1 resembles that of P2 and P3 whereas the fraction of I/SVOCs with higher O:C ratio of P1 is still lower than that of P2 and P3. The fractions of I/SVOC with O:C > 0.3 (0.4, 0.5) are 18.1% (12.1%, 10.8%), 27.8% (18.9%, 16.6%), and 31.0% (21.2%, 19.3%) for P1, P2, and P3 stages, considering that the O:C ratio of the bulk organic species varies from 0.25 to 0.5.

Comparing the EF distribution of I/SVOCs emitted by different types of vehicles under the same driving conditions, as shown in Figure 5, it is clearly demonstrated that the aftertreatment devices favour the formation of I/SVOCs with higher oxidation state. DOC promotes the oxidation of exhaust gases and the organics filtrated by DPF by oxygen and the I/SVOCs with O:C > 0.3 (0.4, 0.5) under the W_cold condition increase from 0.13 to 0.35 (0.11 to 0.18, 0.08 to 0.16) after the equipment of these aftertreatment devices. The respective fractions under W_hot condition increase from 0.18 to 0.26, 0.09 to 0.25, and 0.07 to 0.23, respectively.

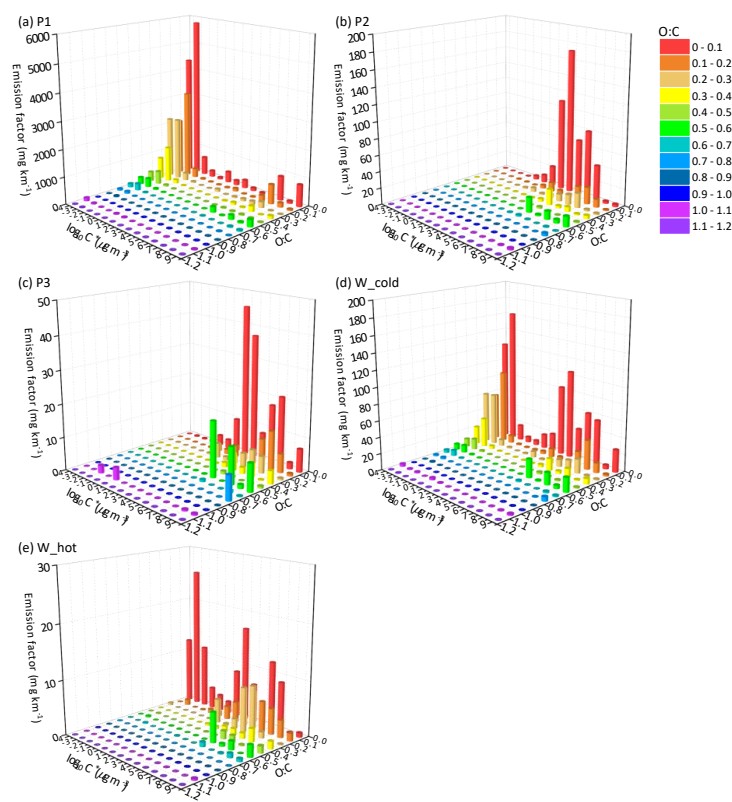

270

**Figure 3. Emission factor distribution of the speciated I/SVOCs of non-(DPF + DOC) vehicles on a 2D-VBS space of (a) low-speed stage (P1), (b) middle-speed stage (P2), (c) high-speed stage (P3), (d) whole (W_cold), and whole (W_hot) driving cycles. Different colours indicate different O:C ratios segmented into 12 bins: 0-0.1, 0.1-0.2, 0.2-0.3, 0.3-0.4, 0.4-0.5, 0.5-0.6, 0.6-0.7, 0.7-0.8, 0.8-0.9, 0.9-1.0, and 1.1-1.2.**



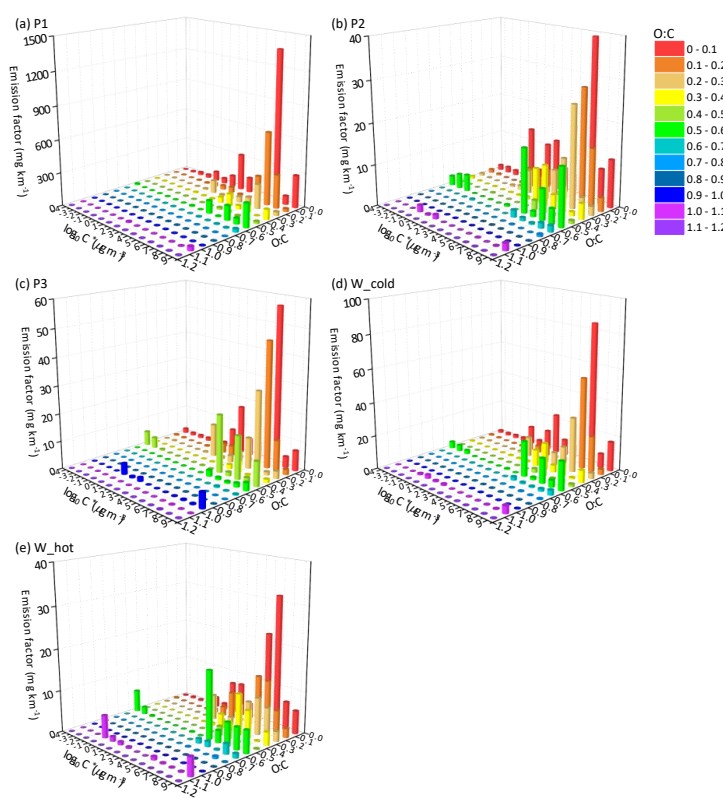

**Figure 4. Emission factor distribution of the speciated I/SVOCs of (DPF + DOC) vehicles on a 2D-VBS space of (a) low-speed stage (P1), (b) middle-speed stage (P2), (c) high-speed stage (P3), (d) whole (W_cold), and whole (W_hot) driving cycles. Different colours indicate different O:C ratios segmented into 12 bins: 0-0.1, 0.1-0.2, 0.2-0.3, 0.3-0.4, 0.4-0.5, 0.5-0.6, 0.6-0.7, 0.7-0.8, 0.8-0.9, 0.9-1.0, and 1.1-1.2.**

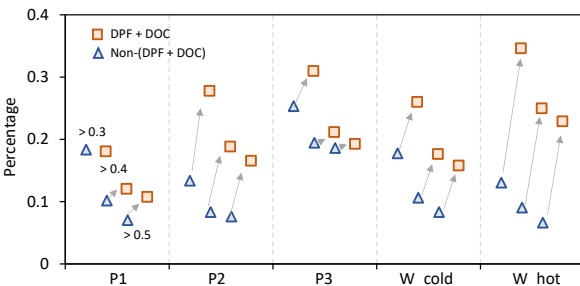

280

**Figure 5. The increment of I/SVOC with O:C > 0.3, 0.4, and 0.5 from non-(DPF + DOC) to (DPF + DOC) vehicles**
**under low-speed stage (P1), middle-speed stage (P2), high-speed stage (P3), and whole (W_cold, W_cold) driving cycles.**

**3.4 Gas particle partitioning of HDDV-emitted I/SVOCs and the uncertainties/artifacts**

Figure 6 shows the g-p partitioning by different compound categories. Generally, the I/SVOCs partition predominantly to the gas phase, with a few exception categories: hopanes, 2ring, 4-ring, and 5-ring PAHs. Distinct patterns are observed between vehicles with and without DPF and DOC. For example, the particle phase contributes 7.4% to the total I/SVOC mass for non-(DPF + DOC) vehicles, whereas it accounts for less than 0.2% for (DPF + DOC) vehicles. Similar mass distributions are observed for alkanes, 3-ring PAHs, and oxygenated species, which confirms the high particle removal efficiency of the aftertreatment devices. The monoaromatic compounds, i.e., $C_2$-$C_6$ alkyl-substituted benzenes, are not detected in the particle phase, regardless of the aftertreatment devices. Over 40% 4-ring PAHs partition to the particle phase for non-(DPF + DOC) vehicles and the portion is reduced to less than 0.1% when DPF and DOC systems are installed. The particle fraction of 2-ring PAHs is 57%, whereas that of 3-ring, 4-ring, and 5-ring PAHs are 8.4%, 43.7%, and 100%, respectively.

The adsorption of gaseous I/SVOCs onto filters causes negative biases in the measured gas phase concentration and positive artifacts in the measured particle phase concentration (Turpin et al., 1994). Compared with quartz filter, which absorbs vapours significantly (May et al., 2013a), teflon has small surface area and is relatively inert. However, the vapor loss to the Teflon surface has long been a concern, especially in smog chamber community (Hu and Kamens, 2007; Mohr et al., 2009). Moreover, the OA concentration in the tailpipe is orders of magnitude higher than that in the ambient air even after the dilution in the CVS system. With such high OA loadings, the g-p partitioning shift to the particle phase. Although inevitable, the bias should be closely watched. For example, the sampling tube is short enough (less than 50 cm) to minimize the g-p conversion in the sampling system (the residence time is less than the time scale to reach g-p equilibrium) (Saleh et al., 2013) and Teflon filter is deployed instead of a quartz filter. Good news is that there will be a significant pressure drop before and after the Teflon filter, and the lower pressure behind the Teflon drives the g-p portioning to the gas phase, which offsets the vapor loss by some extent (Turpin et al., 1994).

We then quantify the sampling artifacts. As shown in Figure 7, the particle mass fraction increases gradually from $\log_{10}C^* = 8$ $\mu$g m$^{-3}$ to $\log_{10}C^* = -4$ $\mu$g m$^{-3}$. Similar trends were observed previously (Lu et al., 2018). There is a peak in the volatility range of $\log_{10}C^* = 3$ to 7 $\mu$g m$^{-3}$ when the particle mass fraction fluctuates around 10% (Figure 7a). The particle fraction decreases to less than 1% between $\log_{10}C^* = -1$ to 2 $\mu$g m$^{-3}$. It is highly likely that the peak reflects the sampling artifacts originated from the vapor loss to the quartz filter. DOC component oxidizes and removes the exhausted gases efficiently, as a consequent of which the sampling artifacts is reduced, i.e., 10% to 1%. The vapor loss occurs in a certain volatility range instead of the whole volatility range, e.g.,





log$_{10}$C$^*$ = 3 to 7 $\mu$g m$^{-3}$ and dominant in log$_{10}$C$^*$ = 5 $\mu$g m$^{-3}$ bin in this study. The gaseous IVOCs in log$_{10}$C$^*$ = 8
$\mu$g m$^{-3}$ bin may be too volatile to be absorbed by quartz filter. For non-(DPF + DOC) vehicles, the particle fraction
approximates 1% at log$_{10}$C$^*$ = 8 $\mu$g m$^{-3}$ and log$_{10}$C$^*$ = 2 $\mu$g m$^{-3}$ bins (the volatility bins adjacent to the bins with
sampling artifacts). If we assume that the particle fraction baseline is 1% during the volatility range of log$_{10}$C$^*$ =
2 to 8 $\mu$g m$^{-3}$, we may deduce that the vapor loss to quartz filter results in a negative bias to the gaseous I/SVOCs
mass with an upper limit of 9% and approximate 90% of the particulate I/SVOCs result from sampling artifacts
in the volatility range of log$_{10}$C$^*$ = 3 to 7 $\mu$g m$^{-3}$. It is also worth mentioning that the absorption bias varies
significantly for different compound categories. For example, substantial 2-ring PAHs are detected in the particle
phase whereas no notable sampling artifacts are observed for phenol benzylic alcohols and benzylic ketone esters.

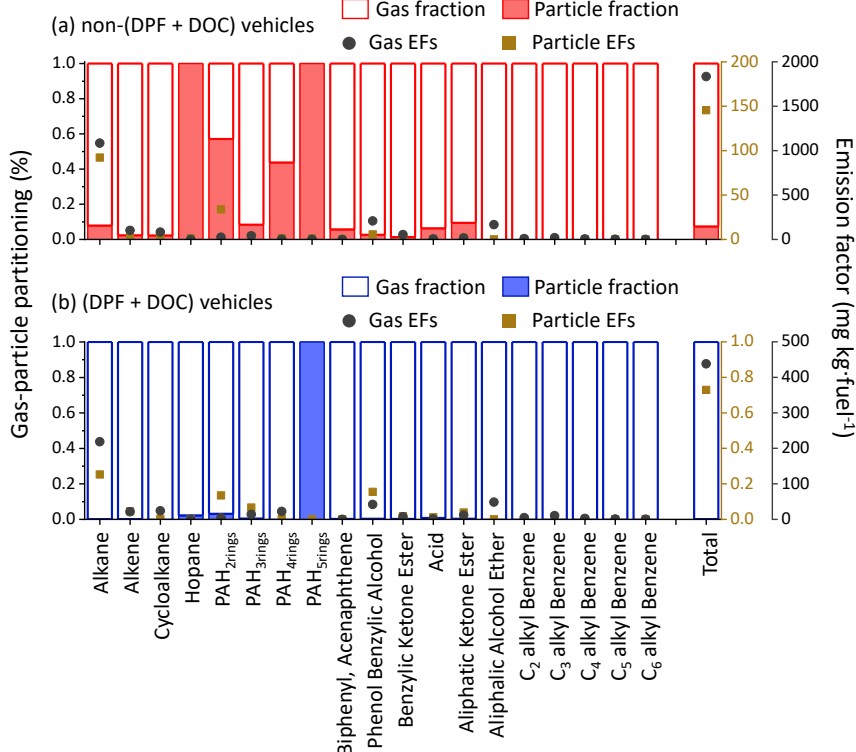


**Figure 6. The gas-particle partitioning of speciated I/SVOCs emitted from (a) non-(DPF + DOC) vehicles and (b) (DPF**
**+ DOC) vehicles. The hollow and filled columns represent the gas and particle fraction, respectively. The grey dots and**
**brown squares represent the emission factors of the gaseous and particulate I/SVOCs.**

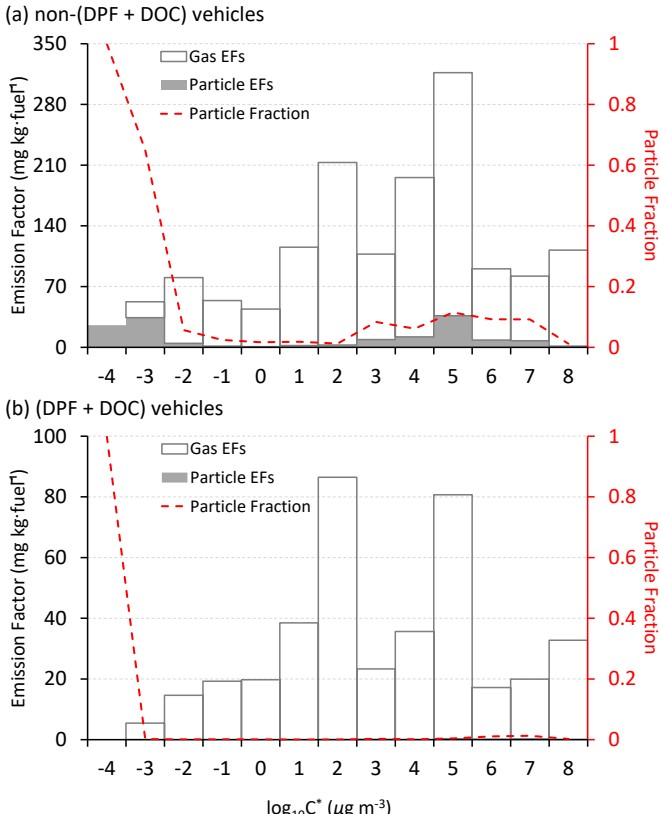


**Figure 7. The emission factors of gaseous and particulate I/SVOCs (hollow and filled stack columns) and the particle mass fraction (red dashed line) in each volatility bin computed from (a) non-(DPF + DOC) vehicles and (b) (DPF + DOC) vehicles.**

## 4. Conclusions

Chassis dynamometer tests of HDDVs complying with multiple emission standards are conducted to characterize the particulate I/SVOCs. Thousands of induvial organic compounds are detected and classified, where alkanes and phenolic compounds are observed to be the most abundant groups. The species-by-species g-p partitioning of the I/SVOCs are discussed separately for vehicles with and without aftertreatment devices. Generally, the I/SVOCs partition to the gas phase dominantly. For non-(DPF + DOC) vehicles, the gaseous I/SVOCs account for ~ 93% of the total mass, except for hopane, $PAH_{4rings}$, and $PAH_{5rings}$. For (DPF + DOC) vehicles, the particulate fraction of I/SVOCs are further reduced to less than 2%. Sampling artifacts of quartz filter absorbing organic vapours are confirmed by the abnormal high signal of 2-ring PAHs, and the uncertainties are discussed thoroughly. Speciation information is highly needed to better predict the thermodynamics of oxidation chemistry. The application of GC × GC-ToF-MS and self-constructed data processing programs achieve the detailed identification and quantification of particulate I/SVOCs. Although not resolved at molecular level, the speciated information enables us to better characterize the emission scenarios and guides the implementation of control



strategies in the future. This approach is versatile and could be applied not only to vehicle emissions but also to
other significant sources prevailing in typical environments, e.g., biomass burning and ship emissions, as well as
ambient samples collected at receptor sites. Putting the speciated I/SVOC data into atmospheric models and
emission inventories, we expect a significantly improved estimation of SOA locally and globally.
**Data availability**
The measurement data used in this study are available in the data repository:
https://figshare.com/articles/dataset/Emission_factor_summary_the_g-
p_partition_and_the_removal_effciency_xlsm/19994603.
**Author contribution:**
Xiao He: Conceptualization, Methodology, Validation, Investigation, Formal Analysis, Writing-Original Draft,
Data Curation, Visualization, Funding Acquisition. Xuan Zheng: Project Management, Validation, Writing-
Review & Editing, Funding Acquisition. Shaojun Zhang: Validation, Writing-Review & Editing. Xuan Wang:
Validation, Writing-Review & Editing. Ting Chen: Investigation. Xiao Zhang: Investigation. Guanghan Huang:
Investigation. Yihuan Cao: Investigation. Liqiang He: Investigation. Xubing Cao: Investigation. Yuan Cheng:
Investigation. Shuxiao Wang: Resources, Writing-Review & Editing, Funding Acquisition. Ye Wu: Resources,
Supervision, Funding Acquisition
**Declaration of Competing Interest**
The authors declare no competing financial interests.
**Acknowledgements**
The authors acknowledge the financial support of the National Natural Science Foundation of China (51978404,
41977180, 42105100, and 22188102) and the Basic Research of Shenzhen Science and Technology Innovation
Commission (JCYJ20190808145218827). The contents of this paper are solely the responsibility of the authors
and do not necessarily represent the official views of the sponsors.



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
