# Peer review of "Comprehensive characterization of particulate the 1"

_Atmospheric Chemistry and Physics, 2022_

## Author Comment (AC1)

I thoroughly reviewed this manuscript. I agree that the subject of the review is very essential. This paper presents the characteristic of the particulate I/SVOCs from chassis dynamometer tests of HDDVs complying with multiple emission standards. Interestingly, this paper provides a versatile approach and could be applied into other significant sources prevailing in typical environments. This may have implications for environmental management. I have listed some specific comments (see below) for this paper, and it is recommended to make revision before publication.

We are grateful for the positive tone in suggesting for accepting this manuscript. Below please see our point-by-point response to the reviewer's comments and suggestions. The comments from the reviewers are in black type and our response texts are marked in blue.

EF and OA in Line 30. These abbreviations have to be explained in their first appearance in the text. Check throughout the manuscript. These are readily recognizable only to the more experienced reader.

Thanks, and revised.

In Introduction, from Line 36 to Line 42, so much dated references are cited, such as: Line 38, Lines 39-40, Line 42. A better presentation is needed along with up-to-date references.

Thanks for the comment. We updated the references as suggested and rephrase the content where appropriate. The following references are adopted:

Sun, S. Y., Zheng, N., Wang, S. J., Li, Y. Y., Hou, S. N., An, Q. R., Chen, C. C., Li, X. Q., Ji, Y. N., and Li, P. Y.: Inhalation Bioaccessibility and Risk Assessment of Metals in PM2.5 Based on a Multiple-Path Particle Dosimetry Model in the Smelting District of Northeast China, Int. J. Env. Res. Public Health, 19, 10.3390/ijerph19158915, 2022b.

Faridi, S., Bayat, R., Cohen, A. J., Sharafkhani, E., Brook, J. R., Niazi, S., Shamsipour, M., Amini, H., Naddafi, K., and Hassanvand, M. S.: Health burden and economic loss attributable to ambient PM2.5 in Iran based on the ground and satellite data, Sci Rep-Uk, 12, 14386, 10.1038/s41598-022-18613-x, 2022.

Nguyen, G. T. H., Nguyen, T. T. T., Shimadera, H., Uranishi, K., Matsuo, T., and Kondo, A.: Estimating Mortality Related to O-3 and PM2.5 under Changing Climate and Emission in Continental Southeast Asia, Aerosol and Air Quality Research, 22, 10.4209/aaqr.220105, 2022.

Li, Y. J., Zhu, Y., Liu, W. J., Yu, S. Y., Tao, S., and Liu, W. X.: Modeling multimedia fate and health risk assessment of polycyclic aromatic hydrocarbons (PAHs) in the coastal regions of the Bohai and Yellow Seas, Sci. Total Environ., 818, 10.1016/j.scitotenv.2021.151789, 2022.

Sun, J., Shen, Z. X., Zhang, T., Kong, S. F., Zhang, H. A., Zhang, Q., Niu, X. Y., Huang, S. S., Xu, H. M., Ho, K. F., and Cao, J. J.: A comprehensive evaluation of PM2.5-bound PAHs and their derivative in winter from six megacities in China: Insight the source-dependent health risk and secondary reactions, Environ. Int., 165, 10.1016/j.envint.2022.107344, 2022a.

Lines 77-78. You expressed "a comprehensive characterization of speciated g-p partition of vehicle emission is yet

to achieve". However, there is no reference, please give some to confirm it.

Thanks, and revised as suggested. The following references are inserted here to support the statement.

Alam, M. S., Zeraati-Rezaei, S., Stark, C. P., Liang, Z. R., Xu, H. M., and Harrison, R. M.: The characterisation of diesel exhaust particles - composition, size distribution and partitioning, Faraday Discuss., 189, 69-84, 10.1039/c5fd00185d, 2016.

Zhao, Y. L., Kreisberg, N. M., Worton, D. R., Isaacman, G., Weber, R. J., Liu, S., Day, D. A., Russell, L. M., Markovic, M. Z., VandenBoer, T. C., Murphy, J. G., Hering, S. V., and Goldstein, A. H.: Insights into Secondary Organic Aerosol Formation Mechanisms from Measured Gas/Particle Partitioning of Specific Organic Tracer Compounds, Environ. Sci. Technol., 47, 3781-3787, 10.1021/es304587x, 2013.

Liu, Y., Gao, Y., Yu, N., Zhang, C., Wang, S., Ma, L., Zhao, J., and Lohmann, R.: Particulate matter, gaseous and particulate polycyclic aromatic hydrocarbons (PAHs) in an urban traffic tunnel of China: Emission from on-road vehicles and gas-particle partitioning, Chemosphere, 134, 52-59, 10.1016/j.chemosphere.2015.03.065, 2015.

The format of the paper still needs to be improved. Such as, Line 80: "integrate" and Line 145: "Nest". Please modify it.

Thanks very much for the comment. We corrected the sentences where appropriate and improve the format of the overall manuscript.

Line 164. You said "wc is the mass fraction of carbon (0.86) in the diesel fuel". How do you get this value 0.86? There is not any analysis about the reason or any reference to support it.

Thanks for the suggestion. According to the national standard: Fuel consumption test methods for heavy-duty commercial vehicles (http://www.catarc.org.cn/upload/201908/20/201908201530245058.pdf), the average molecule formula of diesel fuel is CH1.86. Mass fraction of carbon is 12/13.86 = 0.865. This fraction widely is adopted in previous studies and one of the references is given below: Close T.R. Dallmann, T.W. Kirchstetter, S.J. DeMartini, R.A. Harley Quantifying on-road emissions from gasoline-powered motor vehicles: accounting for the presence of medium- and heavy-duty diesel trucks, Environ. Sci. Technol., 47 (2013), pp. 13873-13881.

Line 173. The sentence "alkane is the most abundant species" is not smooth. Please confirm words "is" and "species" are conflict whether or not?

Thanks for the comments. We checked and confirmed no confliction.

Line 173-175. The two sentences expressed similar idea. It is recommended that they can be combined into one part.

The second sentence is deleted as suggested.

Lines 190-198. This chapter you gave the results about EF. Lines 173-189, the average HDDV-emitted particulate I/SVOCs EFs of cold-start and hot-start driving cycles are expressed, respectively. However, you do not distinguish

the results about cold-start and hot-start in Lines 190-198. Please give an explanation.

When comparing the EFs from cold-start and hot-start, as we did in line 173-189, the EFs from different driving conditions are reported separately. A substantial decrease of EFs from cold-start to cold-start are observed. Further on, we would like to investigate the impacts of aftertreatment devices on the EFs, regardless of the driving conditions. In this regard, for certain vehicle type, i.e., (DOC + DPF) vehicles or non-(DOC + DPF) vehicles, we average the EFs from all the driving conditions and compare the results for different vehicle types.

Line 225. What is the meaning of O-I/SVOCs? You analyzed volatility distribution of I/SVOCs before. What is the difference between O-I/SVOCs and I/SVOCs?

O-I/SVOCs is short for oxygenated intermediate-volatility and semi-volatile organic compounds (I/SVOCs). I/SVOCs are defined on the volatility distribution of organic aerosol and those in the effective saturation concentration ($C^*$) range of $10^{-1}$ to $10^6$ $\mu$g m$^{-3}$. O-I/SVOCs, in particular, refer to the oxygenated fraction of total I/SVOCs. O-I/SVOCs resolved in the particulate I/SVOCs emitted by heavy-duty diesel vehicles (HDDVs) include phenols, benzylic alcohols, aliphatic alcohols, benzylic ketones, acids, and aliphatic ketones, and their respective mass fractions are reported in the manuscript. We missed the full name of O-I/SVOCs in its first appearance in the text and revised in the updated manuscript (line xx).

Lines 266-269.There is some confusion regarding what you refer as the respective fractions. You gave three ratio change results under W_cold condition and W_hot condition. However, what do the three ratio change results correspond to, P1, P2, P3 or may be P3, P2, P1...? Please write it clearly.

Thanks for the suggestion. The CHTC-HT driving cycle (1800 s) is divided into three segments: low-speed (phase one (P1), 342 s), middle-speed 99 (phase two (P2), 988 s), and high-speed (phase three (P3), 470 s) and PM samples are collected separately. Besides, samples are also collected during the whole time (1800 s). 1800s samples collected under cold-start and hot-start conditions are named W_cold and W_hot, where W is short for whole sampling. We add supplementary description of the sampling schedule in the revised manuscript (line xx-xx):

"… (phase three (P3), 470 s) and samples were collected separately. Besides, samples were also collected during the whole sampling time (W, 1800 s) under cold-start and hot-start driving conditions and named W_cold and W_hot for short. Prior…"

Line 309. The same as Q5. The sentence "The particle fraction decreases to less than 1% between log10C* = −1 to 2 μg m-3" is not smooth. Please modify it.

Thanks for the comments. The sentence is rephrased.

Updated text (lines xx-xx):

"…particle fraction decreases to less than 1% between the volatility range of $\log_{10}C^* = -1$ to 2 $\mu$g m$^{-3}$. It is highly likely..."

---

## Author Comment (AC2)

This work collected the particulate intermediate-volatility and semi-volatile organic compounds (I/SVOC) from heavy-duty diesel vehicles (HDDVs) by conducting chassis dynamometer tests. They analyzed the particulate I/SVOCs using two-dimensional gas chromatography time-of-flight mass spectrometry (GC × GC-ToF-MS), speciated the unresolved complex mixtures (UCMs), and reported the emission factor (EF) distribution under diverse driving conditions. Besides, they examined the gas-particle (g-p) partitioning of different compound categories and addressed how the sampling artifacts would affect the quantification of organic compounds. On the overall, I think the paper is worth of publication and meets the interest for the reader of this journal. Suggestions and comments are provided as below:

We thank the reviewer for the positive comments and the recommendation for publication. In the revised manuscript, we rephrase the main text based on the reviewers' suggestions where appropriate. Below please see the point-by-point responses to the individual comments.

1. We know that oxygenated groups, i.e., hydroxyl and carboxylic groups, are not well detected in GC-MS without derivatization with MSTFA. How would you detect those oxygenated compounds without the derivatization and guarantee the recoveries in particular.

Basically, oxygenated groups are not well detected in GC-MS without derivatization especially carboxylic acids when using liquid injection auto sampler. In this way of measuring oxygenated organic compounds, active H atom is substituted with trimethyl-silyl (TMS) reagent, (n-methyl-n-(trimethylsilyl) trifluoroacetamide (MSTFA) (He et al. 2018), as shown in the following figure:

$$M\text{-}OH + \underset{\text{MSTFA}}{CF_3\text{-}C(=O)\text{-}N(CH_3)\text{-}Si(CH_3)_3} \longrightarrow M\text{-}O\text{-}Si(CH_3)_3$$

The derivatization reaction decreases the polarity of the targeted compound and increases the desorption efficiency in the hot inlet liner (usually held at 275°C). In this work, samples are thermally desorbed in an automated thermal desorption system (TD100-xr, Markes International), which is connected to the GC × GC system instead of using a liquid injection auto sampler. Parameters in the thermal desorption system have been throughout optimized to increase the desorption efficiency and the quality control and quality assurance have been well addressed. For details, please see answer to Q5.

2. In section 3.3, when addressing the I/SVOC distribution, the criteria of O:C ratio of 0.3, 0,4, and 0.5 are deployed. Could you explain why the specific numbers are used.

The O:C ratio of the bulk organics with oxygenated functional groups ranges between 0.2 to 0.5 and the that of the highly oxygenated molecules (HOMs) is much higher. For example, the abundant organic compound, terephthalic acid ($C_8H_6O_4$), a significant tracer for anthropogenic emissions (He et al. 2018), has a O:C ratio of 0.5. Besides, it was reported that O/C from ambient urban organic aerosol ranges from 0.2 to 0.8 with a diurnal cycle, and that from

biomass burning OA ranges from 0.3 to 0.4 (Aiken et al. 2008). The average molecular weight of the atmospheric aerosol was reported to be 200 atm and bearing one or two oxygenated functional groups would generate O:C ratio in the particular range.

3. Line 101, it is mentioned that three replicated experiments are conducted. What is the repeatability of the measurements, e.g., reporting the standard deviation. Readers may be interested in the repeatability to confirm the robustness of the methods.

Thanks for the comments. We conducted duplicated experiments for each vehicle and driving conditions. Generally, the repeatability of China VI vehicles is better than that of China IV vehicle.

We now add the standard deviation in the revised manuscript.

Updated text (lines xx-xx, xx-xx):

"…The average HDDV-emitted particulate I/SVOCs EFs of cold-start and hot-start driving cycles are $147.2 \pm 68.3$ and $1.7 \pm 0.3$ mg kg·fuel$^{-1}$ for non-(DPF + DOC) vehicles, $1.6 \pm 0.3$ and $0.9 \pm 0.1$ mg kg·fuel$^{-1}$ for (DPF + DOC) vehicles…

…The EF of alkane derived from non-(DOC + DPF) vehicles under the cold-start condition averages to $92 \pm 42.8$ mg kg·fuel$^{-1}$, which is two order…

…Alkene and cycloalkane show commensurate EFs, with the average values of $2.4 \pm 1.1$ and $1.8 \pm 0.9$ mg kg·fuel$^{-1}$ for cold-start and $0.04 \pm 0.01$ and $0.05 \pm 0.02$ mg kg·fuel$^{-1}$ for hot-start driving…

…2-ring PAHs are detected at abundant concentration of $33.8 \pm 15.7$ mg kg·fuel$^{-1}$, whereas 3-ring, 4-ring, and 5-ring PAHs are detected at much less concentration of $1.5 \pm 0.7$, $1.3 \pm 0.6$, and $0.3 \pm 0.1$ mg kg·fuel$^{-1}$, respectively…"

4. For many categories, such as O-I/SVOCs categories, it seems that they do not have authentic standards. How are these I/SVOCs identified and quantified? The authors should explicitly address the identification of these compound categories in the main text or in the supporting information.

The I/SVOCs emitted by HDDVs are quantified (semi-quantified) by the "three-step" approach. In step one, the compounds of which the authentic standard are available are accurately quantified. However, the portion of accurately quantified mass is relatively small ( < 5%). During step two and three, the I/SVOCs are classified by their mass spectrum pattern and semi-quantified by the surrogates. Taking PAHs$_{4ring}$ as an example, all isomeric 4 ring PAHs exhibit the same fragment pattern: m/z 228 ion has the highest relative intensity and the relative intensity of the m/z 228 fragment ion is three times greater than the ensuing high intensity fragments, i.e., 226, 227, 228. The species are identified into PAHs$_{4ring}$ only if their rules of mass spectrum are in accordance with above-mentioned condition.

For most of the species detected in the GC $\times$ GC chromatogram space, no authentic standard is available. In this case, they are semi-quantified using surrogate standards. For peak X (X denotes any peak that is not quantified) to

be semi-quantified, the first and second retention time are recorded as ($t_{x1}$, $t_{x2}$). The first and second retention times of a series of authentic standards are ($t_{A1}$, $t_{A2}$), ($t_{B1}$, $t_{B2}$), …, and ($t_{N1}$, $t_{N2}$). The distance between peak X and authentic standard A is computed as:

$$s_A = \sqrt[2]{(t_{x1} - t_{A1})^2 + (t_{x2} - t_{A2})^2}$$

Optimal surrogate is the authentic standard of which the distance between peak X and the standard is the minimum of the array ($S_A$, $S_B$, …, and $S_N$). Then the calibration curve of the optimal surrogate is applied to peak X to semi-quantify the compound. The quantification approach is addressed in greater detail in He et al. (2022) and summarized in lines xx-xx.

5. The uncertainty assessment of the emission factors should be presented to convince reader of the quantification method, either in the main text or the supporting information.

Thanks for the valuable suggestion. The uncertainty assessment could be derived from the four aspects. First, the peak area or height of internal standards among different samples are highly reproducible. The repeatability of the internal standard signals reveals that the analytical method could be well duplicated. Second, the calibration curves for all the target compounds, including the hydrocarbons and the functionalized standards, are well established ($R^2$ ranges from 0.950 to 0.999), demonstrating the solidity of this method. Third, the instrument blank tests were conducted throughout the whole experiment, and we did not see observable blank contamination. Duplicated samples were tested randomly, and less than 1% signal intensity of the previous sample was recorded, confirming that the carry-over effect was not a concern. Forth, uncertainties of quantifying individual compounds using surrogates were reported to be 24% and 28.1% (Alam et al. 2018, Huo et al. 2021), which gives an idea of the overall confidential level of the quantification. We give a detailed assessment of the uncertainty of the analytical method and revise the conclusion section based on the reviewer's suggestions.

Updated text in the supporting information (lines xx-xx):

"The uncertainties of the emission factors are assessed. First, the peak area or height of internal standards among different samples are highly reproducible, which reveals that the analytical method could be well duplicated. Second, the calibration curves for all the target compounds are well established. Third, the instrument blank tests were conducted throughout the whole experiment, and no blank contamination is observed. Forth, less than 1% signal intensity of the re-test sample was observed, confirming that the carry-over effect was not a concern. Last, it was reported that using surrogates would introduce un uncertainty of approximate 24% and 28.1%, which gives an overall idea of the uncertainty level of the quantification (Huo et al. 2021, Alam et al. 2018).

6. From Fig 2, it seems like the "phenol benzylic alcohol" is the most abundant O-I/SVOC species. Could you give some example compounds of this category and what are the implications with abundant emission of these compounds?

Some typical "phenol benzylic alcohol" species are phenols with different functional groups as shown in the figure below.

They are a group of aromatic compounds which contribute to secondary organic aerosol (SOA) formation significantly in both gas-phase and aqueous-phase (George et al. 2015, Yee et al. 2013). Their aerosol yields as a function of organic aerosol mass ranges from 0.1-1 (Yee et al. 2013).

7. Line 340, please check throughout the main text and keep "GC × GC" uniform.

Thanks, and revised.

Reference:

Aiken, A. C., P. F. Decarlo, J. H. Kroll, D. R. Worsnop, J. A. Huffman, K. S. Docherty, I. M. Ulbrich, C. Mohr, J. R. Kimmel, D. Sueper, Y. Sun, Q. Zhang, A. Trimborn, M. Northway, P. J. Ziemann, M. R. Canagaratna, T. B. Onasch, M. R. Alfarra, A. S. H. Prevot, J. Dommen, J. Duplissy, A. Metzger, U. Baltensperger & J. L. Jimenez (2008) O/C and OM/OC ratios of primary, secondary, and ambient organic aerosols with high-resolution time-of-flight aerosol mass spectrometry. *Environmental Science & Technology,* 42**,** 4478-4485.

Alam, M. S., S. Zeraati-Rezaei, Z. Liang, C. Stark, H. Xu, A. R. MacKenzie & R. M. Harrison (2018) Mapping and quantifying isomer sets of hydrocarbons ($\geq$ C12) in diesel exhaust, lubricating oil and diesel fuel samples using GC × GC-ToF-MS. *Atmospheric Measurement Techniques,* 11**,** 3047-3058.

George, K. M., T. C. Ruthenburg, J. Smith, L. Yu, Q. Zhang, C. Anastasio & A. M. Dillner (2015) FT-IR quantification of the carbonyl functional group in aqueous-phase secondary organic aerosol from phenols. *Atmospheric Environment,* 100**,** 230-237.

He, X., X. H. H. Huang, K. S. Chow, Q. Wang, T. Zhang, D. Wu & J. Z. Yu (2018) Abundance and Sources of Phthalic Acids, Benzene-Tricarboxylic Acids, and Phenolic Acids in PM2.5 at Urban and Suburban Sites in Southern China. *ACS Earth and Space Chemistry,* 2**,** 147-158.

He, X., X. Zheng, Y. Yan, S. J. Zhang, B. Zhao, X. Wang, G. H. Huang, T. Chen, Y. H. Cao, L. Q. He, X. Chang, S. X. Wang & Y. Wu (2022) Comprehensive chemical characterization of gaseous I/SVOC emissions from heavy-duty diesel vehicles using two-dimensional gas chromatography time-of-flight mass spectrometry. *Environ Pollut,* 305**,** 119284.

Huo, Y., Z. Guo, Y. Liu, D. Wu, X. Ding, Z. Zhao, M. Wu, L. Wang, Y. Feng, Y. Chen, S. Wang, Q. Li & J. Chen (2021) Addressing Unresolved Complex Mixture of I/SVOCs Emitted From Incomplete Combustion of Solid Fuels by Nontarget Analysis. *Journal of Geophysical Research: Atmospheres,* 126.

Yee, L. D., K. E. Kautzman, C. L. Loza, K. A. Schilling, M. M. Coggon, P. S. Chhabra, M. N. Chan, A. W. H. Chan, S. P. Hersey, J. D. Crounse, P. O. Wennberg, R. C. Flagan & J. H. Seinfeld (2013) Secondary organic aerosol formation from biomass burning intermediates: phenol and methoxyphenols. *Atmospheric Chemistry and Physics,* 13**,** 8019-8043.

---

## Author Comment (AC3)

He et al. reported the emission factors of the particulate intermediate and semi-volatile organic compounds by the heavy-duty diesel vehicles, using the two-dimensional gas chromatography time-of-flight mass spectrometry. The authors discussed the volatility distribution of the grouped 21 categories from different driving conditions and discussed the gas and particle phase distribution. The speciated data from the vehicular emission sources is useful for understanding the ambient environmental data, where locational specific source profiles are quite limited. Thus, I would suggest the publication if the author can address my following comments:

We thank the reviewer for the detailed and insightful comments. We provide below a point-to-point response to reviewers' comments. A copy of the manuscript with the changes tracked and a clean copy are submitted together with this response document. The comments from the reviewers are in black type and our response texts are marked in blue.

The abbreviation of ISVOCs and SVOCs in the title should be specified.

Revised as suggested.

Line 30: "EF" should be defined.

Thanks, and revised (line xx).

Line 69: it should be "sub-cooled liquid vapor pressure"

Revised as suggested.

Lines 83-84, the sentence "Particulate I/SVOCs at ascending speed stages are collected and analysed separately" is a repeat to line 79-80, please revise the statement.

Thanks for pointing out the mistake. We delete the repeated sentence.

Line 101: revise "duplicated" to "repeated"

Revised as suggested.

Line 106: repeated words of ""monitored collected".

It should be "monitored collocated" instead of "monitored collected"

Line 117: "Field blank samples are collected collocated at the upstream of the emission pipeline". This sentence is grammatically incorrect, please revise.

Thanks for the comment. We delete collocated as suggested.

Line 120: "A total of 36 filter samples plus 3 field blanks were collected…" please specify the detailed sampling information, e.g., how many samples from the non-(DPF + DOC) vehicles, etc.?

Thanks for comments. We already gave the detailed sampling information in the previous published paper He et al. (2022), Table S2. To make it clear for the readers, we revise the texts.

Updated text (lines xx-xx):

"…A total of 36 filter samples plus 3 field blanks were collected and subjected to the determination of I/SVOCs, among which 1/3 were non-(DPF + DOC) vehicle samples and 2/3 (DPF + DOC) vehicle samples…"

Lines 145 and 146: revise "Nest" to "Next"; "elusion" to "elution".

Thanks very much for pointing out the mistakes. We revised accordingly.

Line 151-155: the specific molecular information of the quantified species should be specified, maybe in the supporting information. For example, what is the carbon range of the measured alkanes, alkenes, cycloalkanes, and hopanes, etc.? The information is necessary as the g-p partitioning are highly associated with the intrinsic chemical properties of the compounds. Besides, the naming of the 4-ring PAHs202 and 4-ring PAHs228 looks weird, and it is not consistent in the main text and in the legend of Figure 2.

We fully agree with the comment and list the molecular information and the formula of the quantified species in the supporting information (Table S1). We classify 4-ring PAHs into two subgroups: 4-ring $PAHs_{202}$ and 4-ring $PAHs_{228}$ considering their different benzene structures and mass spectrum patterns.

| Structure | Molecular weight |
|---|---|
| | 202 |
| | 228 |

We change the names of 4-ring PAHs in the revised manuscript to keep consistency with Figure 2.

Updated text (lines xx-xx):

"…3-ring PAHs, 4-ring PAHs ($PAH_{4rings202}$ and $PAH_{4rings228}$), 5-ring PAHs,…"

There is a negative value of removal efficiency of gas phase data for 4 ring PAHs in Table S2, please check.

Check and revised.

Line 197: what does the "Mono-aromatic compounds" refer to, as it is not within the 21 categories? Is it alkyl benzenes? Please specify.

Monoaromatic compounds are the sum of $C_2$ alkyl benzene, $C_3$ alkyl benzene, $C_4$ alkyl benzene, $C_5$ alkyl benzene, and $C_6$ alkyl benzene, of which the latter ones are resolved within the 21 categories. To avoid ambiguity, we revised the text (lines xx-xx):

"…Mono-aromatic compounds (i.e., $C_2$-$C_6$ alkyl-substituted benzenes), which were measured to take up over 10%..."

Figure 1. I'm confused why there are only six data point for each category, while the sample numbers are 36?

The scatter data points are the averaged values of the emission factor from respective driving conditions and/or test vehicles. To make it clear, we add the description in the revised manuscript (line xx).

"…represent different organic species and driving cycles. The scatter points lying in the left side of the bars represent the averaged emission factors measured from different driving conditions or test vehicles. The square…"

What is the "O-I/SVOCs"? the definition should be given.

Revised as suggested (line xx).

Figure 5, the babel " >0.3, >0.4 and >0.5" should be presented on the x-axis, current format is not clear.

Revised as suggested.

Line 297: "However, the vapor loss to the Teflon surface has long been a concern, especially in smog chamber community". Does the author mean Quartz?

Thanks very much for comment. We miswrite the sentence and delete it.

Lines 298-300: "OA concentration in the tailpipe is orders of magnitude higher than that in the ambient air even after the dilution in the CVS system, which would facilitate partitioning to the particle phase." This statement seems not consistent with the results shown in Figure 6, where most of the measured I/SVOCs are in the gas phase, not in the particle phase.

The freshly emitted OA in the tailpipe is in dynamic portioning between the gas and particle phases and the OA concentration is orders of magnitude higher than that in the ambient environment. According to the Pankow theory, higher concentration of OA facilitates the g-p partitioning into the particle phase (Pankow 1994):

$$f_{part,T} = \left(1 + \frac{1}{k_{OM} \times C_{OA}}\right)^{-1}$$

$$k_{OM} = \frac{RT}{10^6 P_L^0 \delta MW}$$

where R is the gas constant, T is the temperature, $P_L^0$ vapor pressure, MW is the molecular weight of the aerosols, and $\delta$ is the activity coefficient in organic phase.

Even so, I/SVOCs are still measured to be dominated by the gas phase concentration. One of the main reasons is that diesel particulate filter (DPF) removes the diesel emitted PM efficiently (Chen et al. 2021) for (DPF + DOC) vehicles. For non-(DPF + DOC) vehicles, the g-p partitioning varies among different organic compound groups. For example, the alkanes are observed to reside predominantly in the gas phase while 4-ring and 5-ring PAHs in the particle phase.

Has the author measured the total OA concentration? If you did, you can try to model the g-p distribution using

Pankow absorption theory (Pankow, 1994), and to see if it can explain the gas-phase dominant results for the I/SVOCs. And also, why the phase distribution of hopanes is distinctly different between non-(DPF + DOC) vehicles and (DPF + DOC) vehicles, any particular explanations? Reference: Pankow, J. F. (1994). An absorption model of gas/particle partitioning of organic compounds in the atmosphere. Atmospheric Environment, 28(2), 185-188.

We agree with the reviewer that the concentration of total OA is a critical parameter determining the g-p partitioning of I/SVOCs (please see the answers to previous comment where we address the Pankow theory). The monitoring modules connected to the Constant volume sampler (CVS) system include a real-time gas analyzer module (MEXA-7400HLE, HORIBA, Japan) to monitor the transient concentration of CO and $CO_2$, a temperature and relative humidity sensor, an air filtration system, and a flow control system. Total OA concentration is not available in the current design of CVS system. Given this, we directly measure I/SVOC concentration in the gas and particle phase and estimate the g-p particle partitioning of the resolved compound categories instead of modeling the coefficients using Pankow theory.

The distribution of hopane between non-(DPF + DOC) vehicles and (DPF + DOC) vehicles is distinctly different. Hopane, by nature, partitions predominantly into the particle phase and the statement is confirmed by the results obtained from non-(DPF + DOC) vehicles. For (DPF + DOC) vehicles, hopane (or PM) is captured and removed by DPF efficiently, as a result of which the particle phase concentration of hopane is reduced dramatically.

Reference:

Chen, H., X. Sun, X. C. Wang, F. Y. Sun, P. Zhang, L. M. Geng & H. F. Wang (2021) Filtration Efficiency and Regeneration Behavior in a Catalytic Diesel Particulate Filter with the Use of Diesel/Polyoxymethylene Dimethyl Ether Mixture. *Catalysts,* 11.

He, X., X. Zheng, Y. Yan, S. J. Zhang, B. Zhao, X. Wang, G. H. Huang, T. Chen, Y. H. Cao, L. Q. He, X. Chang, S. X. Wang & Y. Wu (2022) Comprehensive chemical characterization of gaseous I/SVOC emissions from heavy-duty diesel vehicles using two-dimensional gas chromatography time-of-flight mass spectrometry. *Environ Pollut,* 305**,** 119284.

Pankow, J. F. (1994) AN ABSORPTION-MODEL OF GAS-PARTICLE PARTITIONING OF ORGANIC-COMPOUNDS IN THE ATMOSPHERE. *Atmospheric Environment,* 28**,** 185-188.

---

## Author Response (AR2)

I thoroughly reviewed this manuscript. I agree that the subject of the review is very essential. This paper presents the characteristic of the particulate I/SVOCs from chassis dynamometer tests of HDDVs complying with multiple emission standards. Interestingly, this paper provides a versatile approach and could be applied into other significant sources prevailing in typical environments. This may have implications for environmental management. I have listed some specific comments (see below) for this paper, and it is recommended to make minor revision before publication.

We thank the reviewer for the detailed comments and the very high mark of this paper. We provide below a point-to-point response to reviewers' comments. A copy of the manuscript with the changes tracked and a clean copy are submitted together with this response document. The comments from the reviewers are in black type and our response texts are marked in blue.

1. The format of the paper still needs to be improved. Such as, Lines 319-321. Please modify it.

Thanks very much for the comment. We use format printer at this place to keep consistency with the other contexts.

2. Lines 100-102, when mentioning the low-speed, middle-speed and high-speed driving cycle, the criteria of 342s, 988s and 470s are deployed. Could you explain why the specific numbers are used.

China heavy-duty commercial vehicle test cycle is a part of the China automotive test cycle (CATC) developed by the China Automotive Technology & Research Center (CATARC). The CHTC is defined by national standard GB/T38146.2, China Automotive Test Cycle Part 2: Heavy-duty Commercial Vehicles, released in October 2019 and applicable from May 2020. The CHTC test replaced the C-WTVC (a modified version of WHVC) for the purpose of vehicle certification.

The CHTC includes six chassis dynamometer driving cycles for various types of heavy commercial vehicles with GVW>3500 kg:

CHTC-B: China heavy-duty commercial vehicle test cycle for city buses

CHTC-C: China heavy-duty commercial vehicle test cycle for inter-city coaches

CHTC-LT: China heavy-duty commercial vehicle test cycle for light trucks of GVW≤5500 kg

CHTC-HT: China heavy-duty commercial vehicle test cycle for heavy trucks of GVW>5500 kg

CHTC-D: China heavy-duty commercial vehicle test cycle for dump trucks

CHTC-TT: China heavy-duty commercial vehicle test cycle for tractor trailers

We adopted CHTC-HT cycle for the heavy-duty vehicles tested in this study. In this particular driving cycle, the speed trace is divided into three segments (low-speed phase 342 s, middle-speed phase 988 s, and high-speed phase 470 s) with increasing vehicle speed to represent the constant speed changes typical of on-road driving.

3. Line 169. What is the unit of $M_{CO2}$, $M_{CO}$, and $M_C$? Please complete it.

Thanks for the suggestion. $M_{CO2}$, $M_{CO}$, and $M_C$ are the molar weight of $CO_2$, $CO$, and $C$ atom, of which the numbers are 44, 28, and 12 g mol$^{-1}$. The sentence is re-written as (line 169):

"$M_{CO2}$, $M_{CO}$, and $M_C$ are the molar weight of $CO_2$ (44 g mol$^{-1}$), $CO$ (28 g mol$^{-1}$), and C (12 g mol$^{-1}$) atom;"